# Relationship between insulin resistance surrogate markers with diabetes and dyslipidemia: A Bayesian network analysis of Korean adults

**Jaeyeop Choi[1], Jonghyun Kim[2], Hyun Sook Oh[3]\***

**1** Department of Applied Statistics, Gachon University, Seongnam-si, Gyeonggi-do, Korea, **2** Department of Applied Statistics, Gachon University, Seongnam-si, Gyeonggi-do, Korea, **3** Department of Applied Statistics, Gachon University, Seongnam-si, Gyeonggi-do, Korea

\* hoh@gachon.ac.kr (HSO)

## Abstract

Insulin resistance (IR) can be optimally assessed using the euglycemic clamp, but practical clinical limitations necessitate surrogate markers. This study leveraged the Bayesian network analysis to evaluate three established IR markers: the Homeostatic Model Assessment of IR (HOMA-IR) using insulin level and fasting blood glucose (FBG), TG-Glucose (TyG) index using triglycerides (TG) and FBG, and TG-to-HDL ratio (TG/HDL ratio) using TG and high-density lipoprotein (HDL), based on the Korean National Health and Nutrition Examination Survey data (2019–2021). Our analysis revealed a sequential association pattern (TG/HDL ratio→TyG index→HOMA-IR), positioning the TyG index as a central connecting marker. The HOMA-IR exhibited strong predictive power for diabetes, while the TG/HDL ratio was most effective for assessing dyslipidemia. However, both had limited crossover utility. In contrast, the TyG index bridged this gap, demonstrating robust predictive capability for both conditions. The Markov blanket analysis illuminated the distinctive metabolic signatures of each marker: The TyG index displayed balanced glucose-lipid metabolic contributions, the HOMA-IR predominantly reflected glucose metabolism and obesity characteristics, and the TG/HDL ratio emphasized lipid metabolism. Notably, the TyG index's predictive performance showed significant enhancement when integrated with obesity information, contrasting with the HOMA-IR's minimal response owing to its inherent incorporation of obesity characteristics. These findings position the TyG index as a superior clinical marker, offering both comprehensive predictive capability and enhanced performance through synergistic integration with obesity measures. While each marker demonstrated reliability, the TyG index's unique combination of versatility and scalability establishes it as an effective tool for comprehensive metabolic risk assessment.

**Data availability statement:** The data used in this study are obtained from the Korean National Health and Nutrition Examination Survey (KNHANES). These data are publicly available and can be accessed through the KNHANES website (http://knhanes.cdc.go.kr).

**Funding:** The author(s) received no specific funding for this work.

**Competing interests:** The authors have declared that no competing interests exist.

## Introduction

Insulin resistance (IR), a condition where tissues show reduced responsiveness to normal insulin levels, fundamentally disrupts glucose and fatty acid regulation [1,2]. This impairment of insulin's role in mediating the transition between nutrient production and storage contributes significantly to chronic conditions such as diabetes and dyslipidemia.

Hyperglycemia damages blood vessels by triggering inflammation, promoting lipid synthesis, and inducing oxidative stress, leading to vascular remodeling and plaque formation [3–5]. These changes increase the risk of complications across multiple organs. Accurate IR assessment is crucial for prevention and management. While the euglycemic clamp is the gold standard for measuring IR, its invasiveness, time demands, and cost limit its clinical use [6–8]. The HOMA-IR offers an alternative approach by evaluating IR through fasting insulin and glucose levels; numerous studies have validated its effectiveness [9–11]. However, the HOMA-IR's utility is constrained by the irregular inclusion of insulin measurements in routine blood work and lack of standardized measurement protocols [12].

To overcome these limitations, researchers have developed non-insulin-based indices, including the TyG index and the TG/HDL ratio. These simpler alternatives have been validated as reliable IR markers, with a few studies suggesting the TyG index may be superior to both the HOMA-IR and the TG/HDL ratio in predicting diabetes and cardiovascular outcomes [13–18]. However, direct comparisons between these three indicators remain scarce.

Despite the TyG index's promise, most research has relied on traditional statistical methods such as logistic regression and Cox proportional hazards models. These approaches, while useful for assessing predictive performance, cannot capture non-linear interactions among variables or fully reflect the complexity of biomedical data [19].

The Bayesian network analysis offers a more comprehensive approach by modeling probabilistic relationships without predefined dependent or explanatory variables [20]. This methodology reveals non-linear interdependencies, which traditional models often overlook, by exploring relationships in an unsupervised manner, while also enabling probability estimations through simulations under various conditional scenarios [20–22]. Furthermore, its visual representation through Directed Acyclic Graphs (DAGs) enhances the interpretation of variable relationships, rendering it particularly valuable in medical and life science applications [23–26].

This study aims to compare the HOMA-IR, TyG index, and TG/HDL ratio as IR surrogate markers for diabetes and dyslipidemia using the Bayesian network analysis. Beyond simple comparison, we explore the pathways through which various clinical and behavioral factors influence these markers. Additionally, we assess their scalability by examining their performance when incorporating key variables such as obesity and hypertension—significant contributors to both diabetes and dyslipidemia. This comprehensive approach provides deeper insights into the markers' predictive capabilities and their broader clinical applicability.

## Materials and methods

### Study design and data extraction

This study analyzed data from the Korea National Health and Nutrition Survey (KNHANES) conducted between 2019 and 2021. The KNHANES, established in 1998 by the Korea Centers for Disease Control and Prevention (KCDC), is an ongoing cross-sectional survey designed to assess and enhance the health and nutritional status of the South Korean population. The survey implements a stratified multistage cluster sampling method to ensure representative sampling of non-institutionalized Koreans aged 1 year and older. Detailed methodology is available on the KNHANES Website. The KNHANES 2019–2021 received approval from the Institutional Review Board of the KCDC, in accordance with the principles outlined in the Declaration of Helsinki. The approval numbers are 2018-01-03-C-A, 2018-01-03-2C-A, and 2018-01-03-5C-A. The requirement for written informed consent was waived by the Institutional Review Board, as the analysis was based on anonymous and de-identified data provided by KNHANES. No additional personal or sensitive data was collected.

The 2019–2021 KNHANES dataset was selected as it represents the most recent period with comprehensive insulin measurements, incorporating health surveys, health examinations, and nutrition surveys. From an initial sample of 22,559 participants, we first excluded 5,390 participants because of incomplete health survey and examination data. We excluded 8,921 participants outside the age range of 40–70 years to focus on the population with highest risk and prevalence of diabetes and dyslipidemia, while avoiding younger individuals with lower incidence rates and elderly subjects with potentially confounding comorbidities [27,28]. Further refinement of the study population involved excluding 726 participants based on their medical history: 366 participants with cardiovascular disease history (including stroke, myocardial infarction, and angina), 277 participants with cancer history (including gastric, liver, colorectal, lung, and thyroid cancers), and 83 participants with kidney disease history. After applying these exclusion criteria, the final study population comprised 8,195 participants (Fig 1).

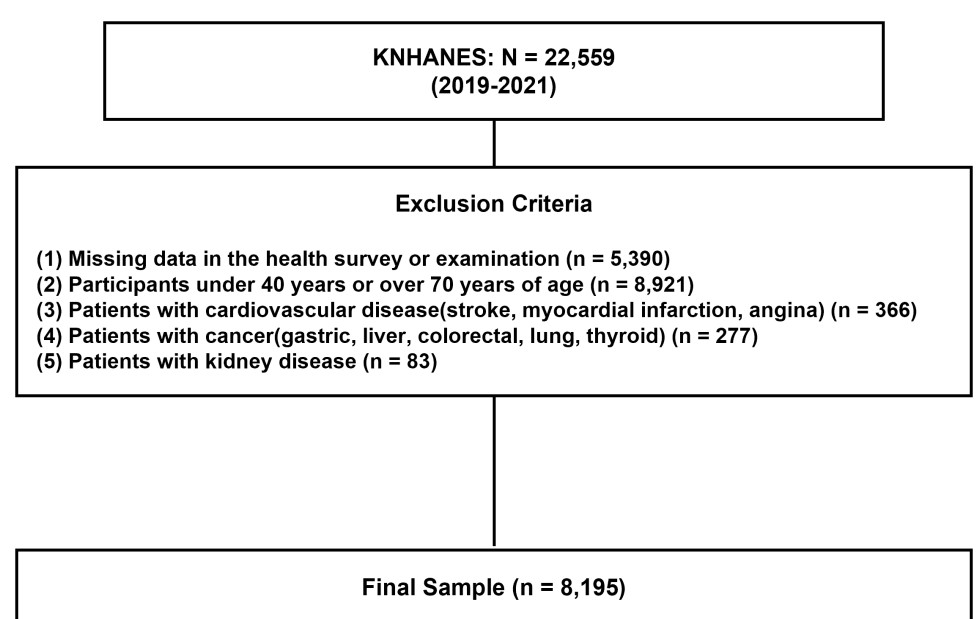

**Fig 1. Flow chart for data selection.** Abbreviation: KNHANES, Korea National Health and Nutrition Survey.

## IR indices

All blood samples were collected after an overnight fast of at least 8 hours, in accordance with the KNHANES protocol. In the KNHANES dataset, insulin levels were measured using the Cobas 8000 e801 analyzer (Roche, Germany) with Elecsys Insulin reagents (Roche, Germany). FBG, TG, and HDL levels were measured using the Labospect 008AS system (Hitachi, Japan). The insulin-based index, the HOMA-IR, was calculated as: HOMA-IR = Insulin (µIU/mL) × FBG (mg/dL)/ 405. Non-insulin-based indices were defined as follows: TyG index = ln[TG (mg/dL) × FBG (mg/dL)/ 2], TG/HDL ratio = TG (mg/dL)/ HDL (mg/dL) [27,28]. Participants were categorized into four groups (Q1–Q4) based on the quartiles of each index.

## Diabetes and dyslipidemia

Diabetes was defined according to the Korean Diabetes Association guidelines as having a diagnosis of diabetes by a physician, FBG ≥ 126 mg/dL, a hemoglobin A1c (HbA1c) ≥6.5%, or a prescription for anti-diabetic medication or insulin therapy [29]. Dyslipidemia was defined according to the Korea Society of Lipid and Atherosclerosis guidelines as meeting any of the following criteria: a diagnosis of dyslipidemia by a physician, a total cholesterol (TCHOL) ≥240 mg/dL, a low-density lipoprotein cholesterol (LDL) ≥160 mg/dL, HDL ≤ 40 mg/dL, TG ≥ 200 mg/dL, or the prescription of medication for dyslipidemia [30].

## Other variables

Socio-demographic factors considered included age, sex, education, and household income. Age was categorized into three groups: 40–49, 50–59, and 60–69 years old. Education level was divided into three categories: Middle school or below, High school, and College or higher. Household income was classified into three groups based on income quartiles as reported in the KNHANES survey: Low/Medium low (lowest and second quartiles), Medium high (third quartile), and High (highest quartile) [31].

Health behaviors included sleep duration, drinking frequency, smoking status, leisure activity, work activity, and muscle activity. Sleep was categorized as < 6 hours or ≥ 6 hours per day, following established thresholds for sleep insufficiency associated with metabolic risk [32]. Drinking was classified as < 1 per month or ≥ 1 per month. Smoking status was classified as no (smoked <5 packs or does not smoke) or yes (smoked ≥5 packs or currently smokes) based on previous epidemiological studies on smoking intensity [33]. Leisure and work activity were categorized as yes or no based on high- or moderate-intensity physical activity at least once a week [34]. Muscle activity was classified as yes if engaging in muscle-strengthening activities at least once a week, and no otherwise.

Clinical variables included systolic blood pressure (SBP), diastolic blood pressure (DBP), FBG, HbA1c, HDL, TG, TCHOL, and LDL. SBP was categorized as < 120 mmHg (Normal), 120–139 mmHg (Caution), and ≥ 140 mmHg (Danger). DBP was classified as < 80 mmHg (Normal), 80–89 mmHg (Caution), and ≥ 90 mmHg (Danger). FBG was categorized as < 100 mg/dL (Normal), 100–125 mg/dL (Caution), and ≥ 126 mg/dL (Danger). HbA1c was divided into <5.7% (Normal), 5.7–6.3% (Caution), and ≥ 6.4% (Danger). HDL was classified as < 40 mg/dL (Danger) and ≥ 40 mg/dL (Normal). TG was categorized as < 150 mg/dL (Normal), 150–199 mg/dL (Caution), and ≥ 200 mg/dL (Danger). TCHOL was classified as < 200 mg/dL (Normal), 200–239 mg/dL (Caution), and ≥ 240 mg/dL (Danger). LDL was classified as < 129 mg/dL (Normal), 130–189 mg/dL (Caution), and ≥ 190 mg/dL (Danger). For unmeasured LDL, Friedewald's formula was appplied: LDL = TCHOL - HDL - (TG/5) [35].

Hypertension was defined according to the Korean Society of Hypertension guidelines as having a diagnosis of hypertension by a physician, SBP ≥ 140 mmHg, DBP ≥ 90 mmHg, or being prescribed anti-hypertensive medication [36]. Obesity was defined using three different indicators: body mass index (BMI), waist circumference (WC), and waist-to-height ratio (WHtR). Participants with BMI ≥ 25 kg/m² were classified as "yes," otherwise "no." Males with WC ≥ 90 cm and

females with WC ≥ 85 cm were classified as "yes," otherwise "no." Participants with WHtR ≥ 0.5 were classified as "yes," otherwise "no" [37,38].

## Statistical analysis

The characteristics of the participants were described based on diabetes and dyslipidemia status, respectively. For each condition, participants were classified into two groups (Yes/No), and comparisons were made to assess variable distributions according to diabetes and dyslipidemia status. Variables were categorized and expressed as weighted percentages, and the Rao-Scott chi-square tests, accounting for survey weights and sampling design, were performed to identify significant differences between groups.

For each IR index, participants were stratified into quartiles: Q1 (≤ first quartile), Q2 (between the first and second quartiles), Q3 (between the second and third quartiles), and Q4 (> third quartile). Distribution across quartile groups was analyzed by diabetes and dyslipidemia status, with tests for significant differences. Statistical significance was defined a P-value < 0.05 in all tests.

Additionally, weighted mutual information was calculated for diabetes and dyslipidemia to assess the mutual dependence of the other variables. The performance of the HOMA-IR, TyG index, and TG/HDL ratio as predictive markers was compared using the receiver operating characteristic (ROC) curves and the area under the curve (AUC), with higher AUC values indicating greater diagnostic performance. Subsequently, a Bayesian network analysis was conducted to explore the relationship structure between all variables.

## Bayesian network

A Bayesian network is a graphical model for probabilistic inference that uses a DAG to represent the joint distribution of variables and their conditional independence relationships [39]. In a DAG, nodes represent random variables $V = \{V_1, V_2, \cdots, V_i\}$, and arcs $A = V \times V$ represent probabilistic dependencies [19,40]. If there is an arc $A_{ij} = \{V_i, V_j\}$ from node $V_i$ to node $V_j$, then $V_i$ is the parent node of $V_j$. The joint probability distribution of variables is:

$$P(V) = \prod_{V_i \in V} P(V_i | \pi(V_i))$$

where $\pi(V_i)$ denotes the set of parent nodes. A Bayesian network $\mathcal{B} = \{G, V\}$ is defined by the graph, $G = (V, A)$, and the probability distribution, $P(V)$ [41].

This study uses DAG analysis to reveal complex relationships between IR markers and metabolic disorders. Unlike traditional statistical methods, DAGs model probabilistic relationships, visualizing direct and indirect connections. This approach illuminates the interplay between IR markers and their relationships with diabetes, dyslipidemia, and other clinical and behavioral factors.

Additionally, the Markov blanket of a node $V_i$ includes its parents, children, and the parents of its children. It is the minimal set of information required to explain $V_i$ while making all other variables conditionally independent [42–44].

In this study, the Markov blanket analysis identifies variables with direct probabilistic relationships to each IR measure, filtering out indirect connections. This approach determines the minimal set of variables that provide complete information about each index's relationships within the network, clarifying how these IR measures relate to clinical factors and to each other in the metabolic structure.

There are three main methods for network structure learning: Score-based learning, Constraint-based learning, and Hybrid learning [45]. Score-based learning finds the graph with the highest objective function, using scores such as the Bayesian score or Information-theoretic score. Constraint-based learning uses conditional independence tests to build the graph. Hybrid learning combines both approaches by restricting the search space through tests and selecting the highest-scoring graph [46].

In this study, network structure learning was conducted using the score-based Hill-climbing algorithm with the Akaike Information Criterion (AIC) as the score. To enhance model robustness, 1,000 bootstrap iterations were performed, and the network structure was refined by including only arcs that appeared in at least 75% of the iterations through model averaging [47-48]. Conditional probability estimation was performed using the "cpquery" function from the "bnlearn" package. All analyses were performed using R version 4.3.2 and the "bnlearn" package version 4.9.1.

## Results

### General characteristics of the study population

The study sample characteristics was summarized by diabetes and dyslipidemia status (Table 1). Diabetes prevalence was 15.20%, and dyslipidemia was 51.86%. While the overall sex ratio was balanced, males were more prevalent in both groups. Among diabetes patients, frequency increased with age, peaking in the 40s, followed by the 50s and 60s. Dyslipidemia was most common in the 40s, followed by the 60s and 50s.

The Rao-Scott tests showed that most variables, except sleep duration, drinking frequency, and work activity, were significantly associated with diabetes or dyslipidemia. Obesity (BMI) rates were 38.62% overall, 58.14% in diabetes, and 48.31% in dyslipidemia. Hypertension prevalence was 32.51% overall, 54.67% in diabetes, and 41.61% in dyslipidemia.

### Distribution of IR indices across quartile groups

Table 2 shows participant distribution across IR index quartiles, stratified by diabetes and dyslipidemia status. Patients with diabetes showed high IR markers, with the majority falling in the highest quartile group (Q4 or Q3) across all indices. Specifically, for TyG index, 52.66% were in Q4 and 25.53% in Q3. HOMA-IR values were in Q4 at 58.09%, with 21.21% in Q3. The TG/HDL ratio showed 38.28% in Q4 and 28.90% in Q3.

Among dyslipidemia patients, a similar pattern emerged but with slightly different distribution. For TyG index, 42.81% were in Q4 and 25.71% in Q3. HOMA-IR measurements showed 35.48% in Q4 and 28.10% in Q3. The TG/HDL ratio showed 44.48% in Q4 and 24.84% in Q3.

### Bayesian network analysis

For diabetes, the HOMA-IR demonstrated the highest predictive performance with both the greatest mutual information (5.11%) and AUC (0.76), followed by the TyG index (4.25%, 0.74) and TG/HDL ratio (1.92%, 0.64) (Figs 2A and 2B). Conversely, for dyslipidemia, the TG/HDL ratio showed the best performance, ranking highest in both mutual information (14.01%) and AUC (0.78), followed by the TyG index (12.05%, 0.77) and the HOMA-IR (5.05%, 0.69) (Figs 2C and 2D).

The DAG structure of the BN model is illustrated in Fig 3. Arc thickness represents AIC values, with thicker arcs indicating stronger connections. The association between the TG/HDL ratio and TyG index was stronger than that between the TyG index and HOMA-IR. Marginal probabilities of each node are provided in the supplementary materials (S1 Fig).

All three IR indices shared FBG, a child node of diabetes, as a parent node. These indices formed a sequential connection structure flowing from TG/HDL ratio through TyG index to HOMA-IR (Fig 3) [46]. The Markov blanket analysis revealed that the TyG index was included in the Markov blankets of both HOMA-IR and TG/HDL ratio, while HOMA-IR and TG/HDL ratio did not appear in each other's Markov blankets (Fig 4).

Specifically, the Markov blanket of the HOMA-IR included the TyG index and FBG, along with age and obesity indicators such as BMI, WC, and WHtR. The Markov blanket of the TyG index included the HOMA-IR, TG/HDL ratio, FBG, as well as Sex, Age, Frequency of drinking, History of smoking, TG, and HDL. The Markov blanket of the TG/HDL ratio consisted of the TyG index and FBG, as well as Sex, Education, Muscle activity, TG, and HDL.

**Table 1. Summary of study participants by diabetes and dyslipidemia.**

| Variables | Levels | Total | Diabetes | | | Dyslipidemia | | |
|---|---|---|---|---|---|---|---|---|
| | | N = 8,195 | No | Yes | p-value | No | Yes | p-value |
| | | | 6,882 (84.80%) | 1,313 (15.20%) | | 3,916 (48.14%) | 4,279 (51.86%) | |
| Sex | M | 3,496 (49.61%) | 2,784 (47.44%) | 712 (61.69%) | < 0.01 | 1,436 (42.19%) | 2,060 (56.50%) | < 0.01 |
| Age | 40-49 | 2,713 (37.21%) | 2,482 (39.98%) | 231 (21.83%) | < 0.01 | 1,639 (45.06%) | 1,074 (29.94%) | < 0.01 |
| | 50-59 | 2,809 (36.90%) | 2,389 (36.75%) | 420 (37.69%) | | 1,294 (35.05%) | 1,515 (38.61%) | |
| | 60-69 | 2,673 (25.89%) | 2,011 (23.27%) | 662 (40.48%) | | 983 (19.89%) | 1,690 (31.45%) | |
| Education | ≤Middle school | 1,944 (19.39%) | 1,442 (17.23%) | 502 (31.45%) | < 0.01 | 706 (14.87%) | 1,238 (23.58%) | < 0.01 |
| | High School | 3,088 (39.22%) | 2,623 (39.39%) | 465 (38.27%) | | 1,525 (39.88%) | 1,563 (38.60%) | |
| | ≥College | 3,163 (41.39%) | 2,817 (43.38%) | 346 (30.28%) | | 1,685 (45.25%) | 1,478 (37.82%) | |
| House incm | Low&Medium low | 2,934 (33.20%) | 2,345 (31.74%) | 589 (41.34%) | < 0.01 | 1,278 (30.84%) | 1,656 (35.38%) | < 0.01 |
| | Medium high | 2,438 (30.84%) | 2,072 (31.16%) | 366 (29.08%) | | 1,214 (31.70%) | 1,224 (30.05%) | |
| | High | 2,823 (35.96%) | 2,465 (37.10%) | 358 (29.58%) | | 1,424 (37.46%) | 1,399 (34.57%) | |
| Sleep duration | < 6 hours | 4,699 (56.88%) | 3,970 (57.25%) | 729 (54.82%) | 0.20 | 2,281 (57.45%) | 2,418 (56.35%) | 0.39 |
| Frequency of drinking | ≥1 per month | 4,322 (55.34%) | 3,648 (55.27%) | 674 (55.77%) | 0.77 | 2,134 (55.83%) | 2,188 (54.89%) | 0.45 |
| History of smoking | Yes | 1,430 (19.46%) | 1,125 (18.33%) | 305 (25.78%) | < 0.01 | 568 (15.91%) | 862 (22.76%) | < 0.01 |
| Leisure activity | Yes | 2,406 (31.05%) | 2,104 (32.32%) | 302 (24.00%) | < 0.01 | 1,208 (32.54%) | 1,198 (29.67%) | 0.02 |
| Work activity | Yes | 553 (7.31%) | 480 (7.53%) | 73 (6.12%) | 0.13 | 272 (7.28%) | 281 (7.35%) | 0.92 |
| Muscle activity | Yes | 2,031 (25.84%) | 1,742 (26.50%) | 289 (22.16%) | 0.01 | 999 (26.66%) | 1,032 (25.07%) | 0.15 |
| BMI | Yes | 3,099 (38.62%) | 2,348 (35.12%) | 751 (58.14%) | < 0.01 | 1,099 (28.18%) | 2,000 (48.31%) | < 0.01 |
| WC | Yes | 3,089 (37.46%) | 2,290 (33.33%) | 799 (60.49%) | < 0.01 | 1,040 (26.18%) | 2,049 (47.92%) | < 0.01 |
| WHtR | Yes | 5,036 (60.72%) | 3,940 (56.78%) | 1,096 (82.70%) | < 0.01 | 1,890 (47.45%) | 3,146 (73.04%) | < 0.01 |
| Hypertension | Yes | 2,782 (32.51%) | 2,045 (28.54%) | 737 (54.67%) | < 0.01 | 931 (22.71%) | 1,851 (41.61%) | < 0.01 |
| SBP | Normal | 4,325 (53.72%) | 3,823 (56.27%) | 502 (39.54%) | < 0.01 | 2,359 (61.17%) | 1,966 (46.81%) | < 0.01 |
| | Caution | 2,978 (35.96%) | 2,347 (33.97%) | 631 (47.06%) | | 1,213 (30.28%) | 1,765 (41.23%) | |
| | Danger | 892 (10.32%) | 712 (9.76%) | 180 (13.40%) | | 344 (8.55%) | 548 (11.96%) | |
| DBP | Normal | 5,054 (60.20%) | 4,242 (60.31%) | 812 (59.61%) | 0.90 | 2,557 (64.97%) | 2,497 (55.77%) | < 0.01 |
| | Caution | 2,319 (29.05%) | 1,950 (29.00%) | 369 (29.28%) | | 1,010 (25.93%) | 1,309 (31.94%) | |
| | Danger | 822 (10.75%) | 690 (10.69%) | 132 (11.11%) | | 349 (9.10%) | 473 (12.29%) | |
| FBG | Normal | 4,565 (56.06%) | 4,492 (65.28%) | 73 (4.64%) | < 0.01 | 2,584 (66.82%) | 1,981 (46.08%) | < 0.01 |
| | Caution | 2,887 (35.06%) | 2,390 (34.72%) | 497 (36.94%) | | 1,143 (28.60%) | 1,744 (41.05%) | |
| | Danger | 743 (8.88%) | 0 (0.00%) | 743 (58.42%) | | 189 (4.58%) | 554 (12.87%) | |
| HbA1c | Normal | 3,742 (47.47%) | 3,714 (55.64%) | 28 (1.85%) | < 0.01 | 2,264 (59.37%) | 1,478 (36.42%) | < 0.01 |
| | Caution | 3,449 (40.73%) | 3,168 (44.36%) | 281 (20.49%) | | 1,404 (34.66%) | 2,045 (46.36%) | |
| | Danger | 1,004 (11.80%) | 0 (0.00%) | 1,004 (77.66%) | | 248 (5.97%) | 756 (17.22%) | |
| TG | Normal | 5,636 (67.22%) | 4,915 (69.82%) | 721 (52.70%) | < 0.01 | 3,432 (87.25%) | 2,204 (48.62%) | < 0.01 |
| | Caution | 1,226 (15.13%) | 961 (14.16%) | 265 (20.54%) | | 480 (12.68%) | 746 (17.41%) | |
| | Danger | 1,333 (17.65%) | 1,006 (16.02%) | 327 (26.76%) | | 4 (0.07%) | 1,329 (33.97%) | |
| HDL | Danger | 1,235 (15.91%) | 909 (13.91%) | 326 (27.04%) | < 0.01 | 0 (0.00%) | 1,235 (30.67%) | < 0.01 |
| LDL | Normal | 5,127 (61.78%) | 4,102 (59.13%) | 1,025 (76.54%) | < 0.01 | 2,577 (65.09%) | 2,550 (58.71%) | < 0.01 |
| | Caution | 2,083 (26.05%) | 1,890 (27.88%) | 193 (15.84%) | | 1,339 (34.91%) | 744 (17.82%) | |
| | Danger | 985 (12.17%) | 890 (12.99%) | 95 (7.62%) | | 0 (0.00%) | 985 (23.47%) | |

*(Continued)*

**Table 1.** (Continued)

| Variables | Levels | Total | Diabetes | | | Dyslipidemia | | |
|---|---|---|---|---|---|---|---|---|
| | | N = 8,195 | No | Yes | p-value | No | Yes | p-value |
| | | | 6,882 (84.80%) | 1,313 (15.20%) | | 3,916 (48.14%) | 4,279 (51.86%) | |
| TCHOL | Normal | 4,344 (52.29%) | 3,439 (49.52%) | 905 (67.72%) | < 0.01 | 2,205 (56.44%) | 2,139 (48.43%) | < 0.01 |
| | Caution | 2,729 (33.83%) | 2,449 (35.99%) | 280 (21.78%) | | 1,711 (43.56%) | 1,018 (24.80%) | |
| | Danger | 1,122 (13.88%) | 994 (14.49%) | 128 (10.50%) | | 0 (0.00%) | 1,122 (26.77%) | |
| Dyslipidemia | Yes | 4,279 (51.86%) | 3,302 (47.80%) | 977 (74.52%) | < 0.01 | – | – | – |
| Diabetes | Yes | 1,313 (15.20%) | – | – | – | 336 (8.04%) | 977 (21.84%) | |

Data was presented as n (raw frequency) with weighted proportion in parentheses. P-values were obtained using the Rao-Scott chi-square tests. Abbreviations: BMI, body mass index; WC, waist circumference; WHtR, waist-to-height ratio; SBP, systolic blood pressure; DBP, diastolic blood pressure; FBG, fasting blood glucose; HbA1c, hemoglobin A1c; TG, triglycerides; HDL, high-density lipoprotein; LDL, low-density lipoprotein; TCHOL, total cholesterol.

**Table 2.** Summary of IR indices across quartiles.

| Variables | Levels | Diabetes | | | Dyslipidemia | | |
|---|---|---|---|---|---|---|---|
| | | No | Yes | p-value | No | Yes | p-value |
| | | 6,882 (84.80%) | 1,313 (15.20%) | | 3,916 (48.14%) | 4,279 (51.86%) | |
| TyG | Q1 (≤8.22) | 2,032 (28.30%) | 93 (6.52%) | < 0.01 | 1,524 (38.11%) | 601 (12.81%) | < 0.01 |
| | Q2 (8.22–8.66) | 1,886 (26.77%) | 209 (15.29%) | | 1,242 (31.87%) | 853 (18.67%) | |
| | Q3 (8.66–9.11) | 1,688 (24.88%) | 351 (25.53%) | | 922 (24.20%) | 1,117 (25.71%) | |
| | Q4 (>9.11) | 1,276 (20.05%) | 660 (52.66%) | | 228 (5.82%) | 1,708 (42.81%) | |
| TG/HDL | Q1 (≤1.37) | 1,991 (27.47%) | 168 (11.31%) | < 0.01 | 1,512 (37.44%) | 647 (13.48%) | < 0.01 |
| | Q2 (1.37–2.27) | 1,811 (25.62%) | 286 (21.51%) | | 1,295 (33.39%) | 802 (17.20%) | |
| | Q3 (2.27–3.83) | 1,633 (24.28%) | 383 (28.90%) | | 959 (25.14%) | 1,057 (24.84%) | |
| | Q4 (>3.83) | 1,447 (22.63%) | 476 (38.28%) | | 150 (4.03%) | 1,773 (44.48%) | |
| HOMA-IR | Q1 (≤1.15) | 1,946 (28.08%) | 106 (7.98%) | < 0.01 | 1,365 (35.50%) | 687 (15.29%) | < 0.01 |
| | Q2 (1.15–1.75) | 1,887 (27.10%) | 175 (12.72%) | | 1,128 (29.00%) | 934 (21.13%) | |
| | Q3 (1.75–2.79) | 1,774 (25.74%) | 296 (21.21%) | | 867 (21.77%) | 1,203 (28.10%) | |
| | Q4 (>2.79) | 1,275 (19.08%) | 736 (58.09%) | | 556 (13.73%) | 1,455 (35.48%) | |

P-values were obtained using the Rao-Scott chi-square tests. Abbreviations: TyG, triglyceride-glucose index; HOMA-IR, homeostasis model assessment of insulin resistance; TG/HDL, triglyceride-to-high-density lipoprotein cholesterol ratio; Q1, 1st quartile group; Q2, 2nd quartile group; Q3, 3rd quartile group; Q4, 4th quartile group.

So, Obesity indicators (BMI, WC, WHtR) were only part of the Markov blanket of the HOMA-IR and were not included in the Markov blankets of the other indices. Meanwhile, FBG acted as a d-separator for the three indices, reinforcing its role as a diagnostic marker for diabetes, with the strongest association observed with the TyG index (Fig. 3) [46].

For dyslipidemia, the lipid clinical variables TG, LDL, HDL, TCHOL, and Age were identified as parent nodes. Among these, HDL had the highest association with the TG/HDL ratio, while TG, which exhibited the highest mutual information with dyslipidemia, showed the strongest association with the TyG index (Figs 2C and 3).

## Conditional probabilities

The conditional probabilities of diabetes for the HOMA-IR across quartiles Q1 to Q4 were 0.06, 0.09, 0.16, and 0.34, respectively. For the TyG index, the probabilities were 0.05, 0.10, 0.17, and 0.35. For the TG/HDL ratio, the probabilities were 0.08, 0.12, 0.19, and 0.25. These values are shown in Fig 5A.

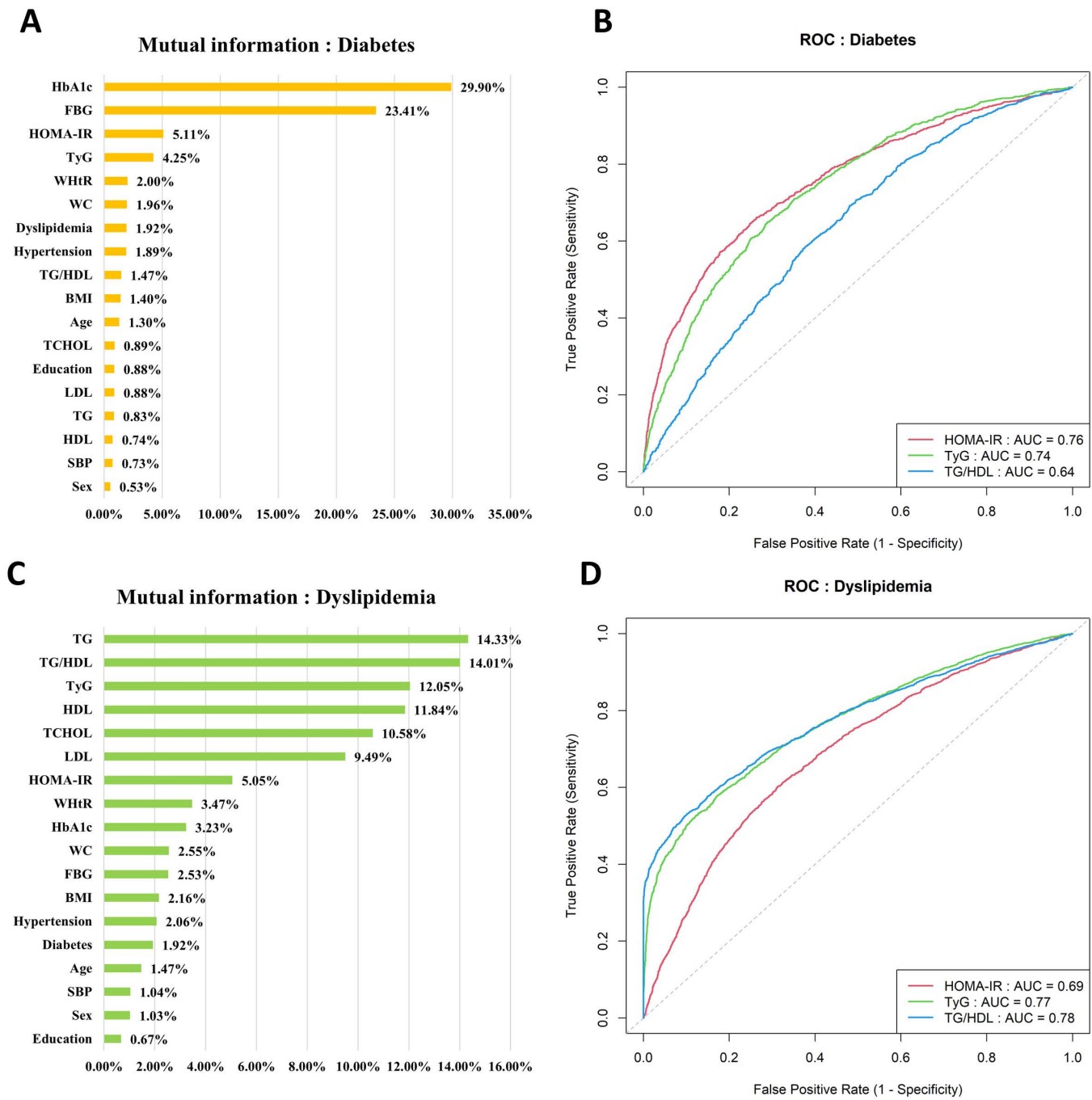

**Fig 2. Mutual information and ROC curve of diabetes (A, B) and dyslipidemia (C, D).** Abbreviation: ROC, receiver operating characteristic; AUC, area under the curve. BMI, body mass index; WC, waist circumference; WHtR, waist-to-height ratio; SBP, systolic blood pressure; FBG, fasting blood glucose; HbA1c, hemoglobin A1c; TG, triglycerides; HDL, high-density lipoprotein; LDL, low-density lipoprotein; TCHOL, total cholesterol; HOMA-IR, homeostasis model assessment of insulin resistance; TyG index, triglyceride-glucose index; TG/HDL, triglyceride-to-high-density lipoprotein cholesterol ratio.

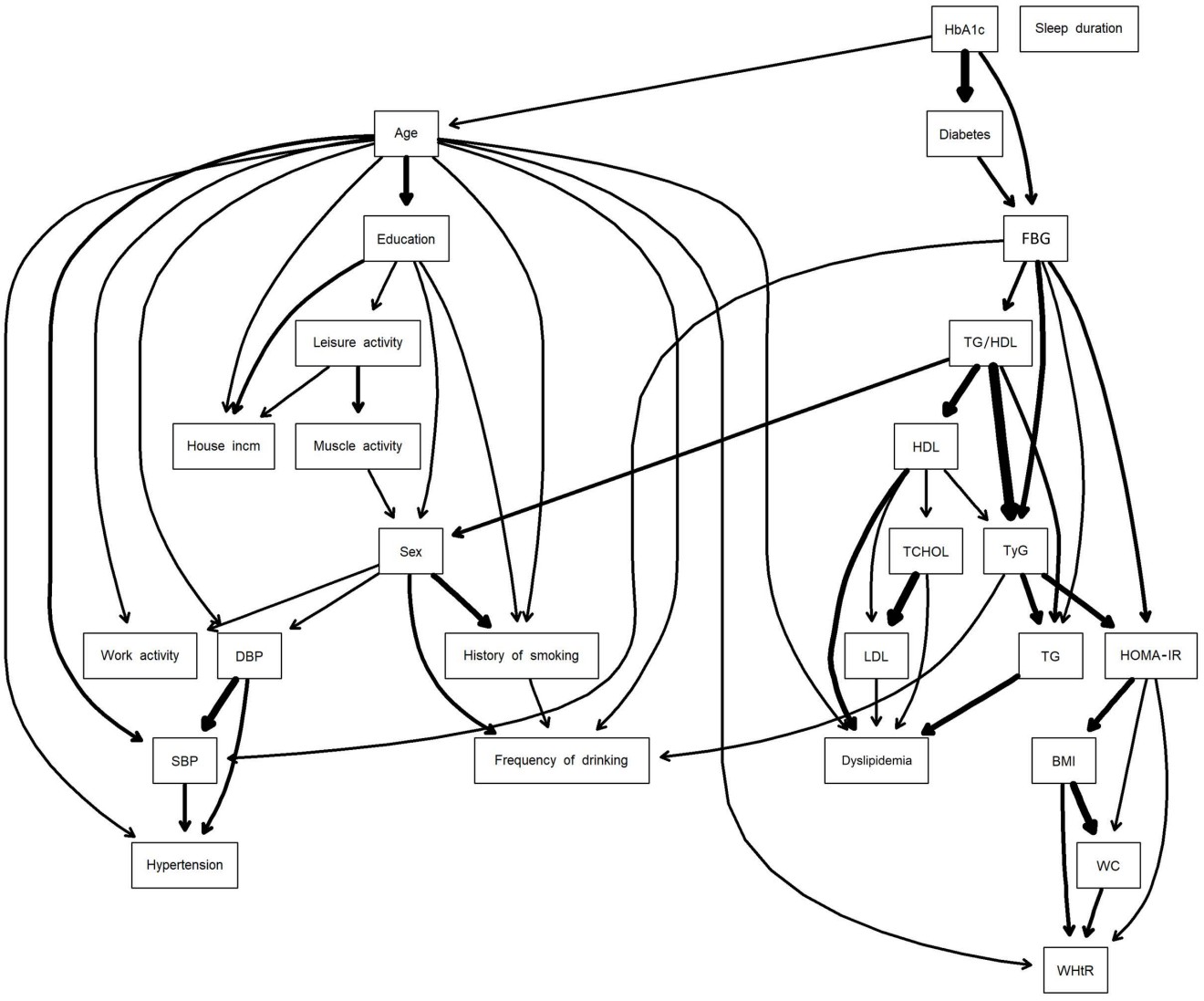

**Fig 3. A directed acyclic graph of Bayesian network obtained through hill-climbing learning algorithm with threshold = 0.75 by model averaging over 1,000 networks.** Arc thickness represents the strength of association based on the Akaike Information Criterion (AIC). Abbreviation: BMI, body mass index; WC, waist circumference; WHtR, waist-to-height ratio; SBP, systolic blood pressure; FBG, fasting blood glucose; HbA1c, hemoglobin A1c; TG, triglycerides; HDL, high-density lipoprotein; LDL, low-density lipoprotein; TCHOL, total cholesterol; HOMA-IR, homeostasis model assessment of insulin resistance; TyG index, triglyceride-glucose index; TG/HDL ratio, triglyceride-to-high-density lipoprotein cholesterol ratio.

Similarly, the conditional probabilities of dyslipidemia for the HOMA-IR across quartiles Q1 to Q4 were 0.43, 0.48, 0.56, and 0.64, respectively. For the TyG index, the probabilities were 0.36, 0.40, 0.49, and 0.87. For the TG/HDL ratio, the probabilities were 0.36, 0.38, 0.48, and 0.92. These values are shown in Fig 5B.

When additional information on obesity (based on BMI) was provided, the conditional probability of diabetes across quartiles of the TyG index was as illustrated in Fig 6A. For the TyG index in Q4, if the individual was non-obese (BMI = "no"), the conditional probability of diabetes was 0.29. However, when the additional information indicated obesity

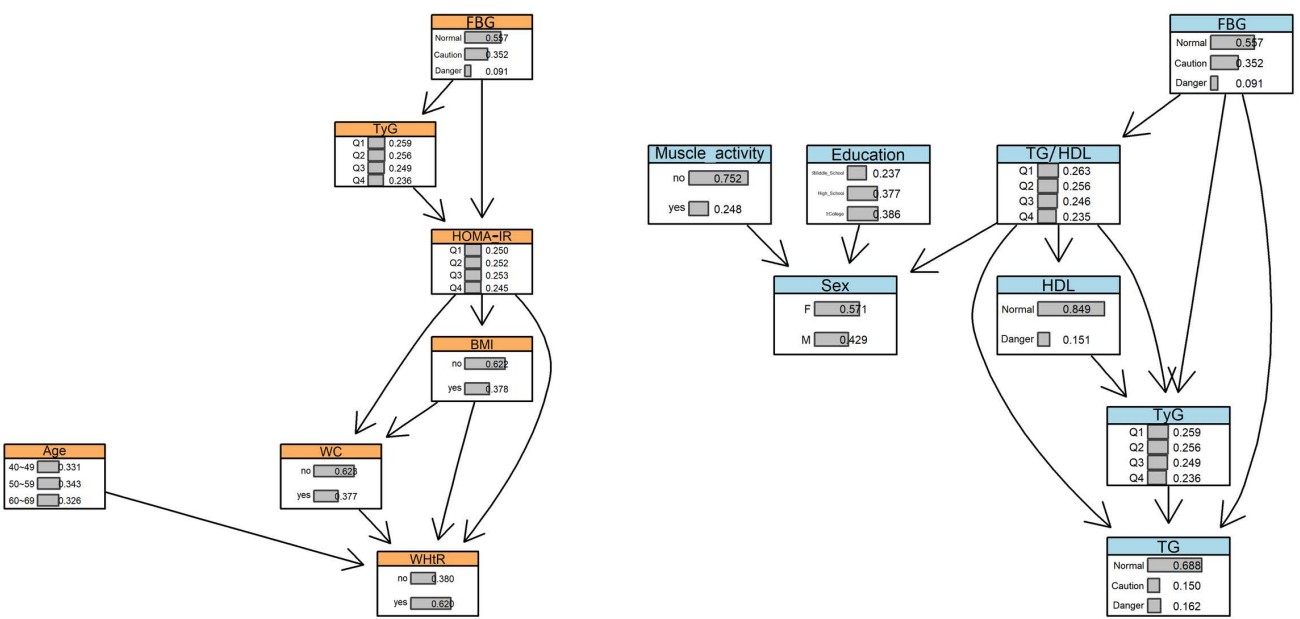

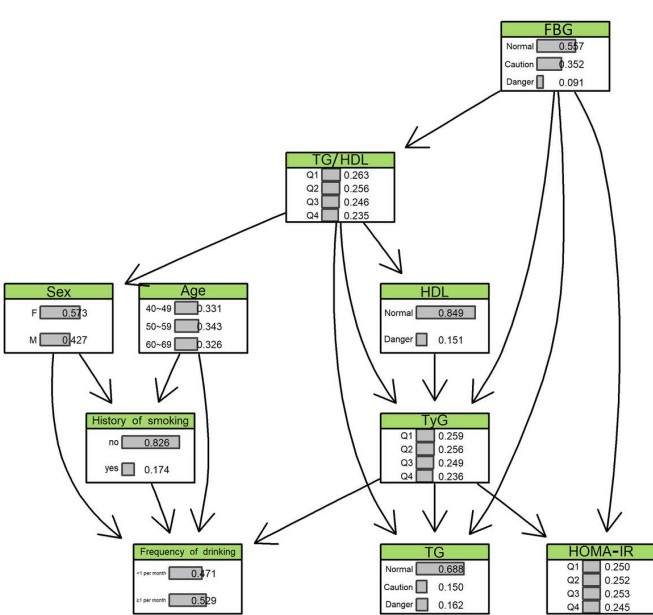

**Fig 4. Markov blankets of HOMA-IR (A), TG/HDL ratio (B), and TyG index (C).** Abbreviation: BMI, body mass index; WC, waist circumference; WHtR, waist-to-height ratio; SBP, systolic blood pressure; FBG, fasting blood glucose; HbA1c, hemoglobin A1c; TG, triglycerides; HDL, high-density lipoprotein; LDL, low-density lipoprotein; TCHOL, total cholesterol; HOMA-IR, homeostasis model assessment of insulin resistance; TyG index, triglyceride-glucose index; TG/HDL, triglyceride-to-high-density lipoprotein cholesterol ratio.

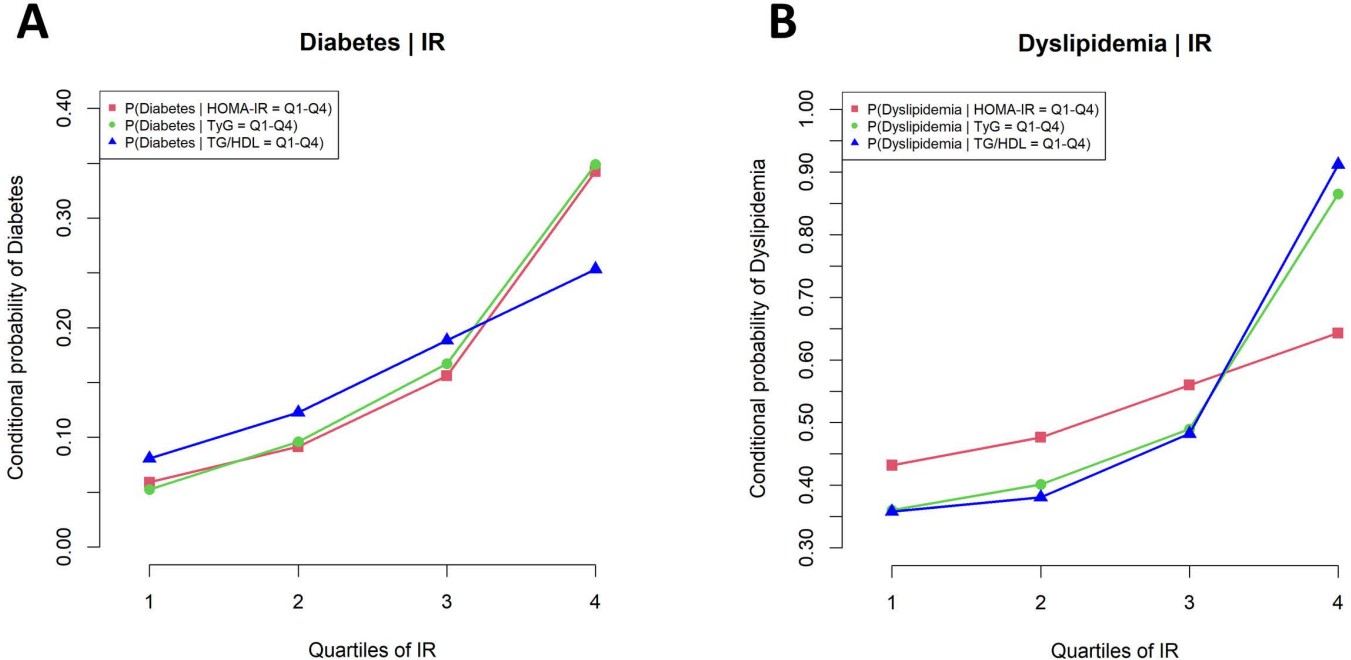

**Fig 5. Conditional probabilities of diabetes (A) and dyslipidemia (B) across the quartile groups (Q1-Q4) of HOMA-IR, TyG index, and TG/HDL ratio.** Abbreviation: HOMA-IR, homeostasis model assessment of insulin resistance; TyG index, triglyceride-glucose index; TG/HDL, triglyceride-to-high-density lipoprotein cholesterol ratio; Q1, 1st quartile group; Q2, 2nd quartile group; Q3, 3rd quartile group; Q4, 4th quartile group.

(BMI = "yes"), the probability increased to 0.39, demonstrating the change in diabetes probability due to the inclusion of obesity information.

However, for the HOMA-IR in Q4, the conditional probability of diabetes was similar between non-obese (BMI = "no", 0.35) and obese (BMI = "yes", 0.34) individuals (Fig 6B), suggesting that the diabetes risk associated with high HOMA-IR values was independent of obesity status.

When additional information on hypertension was provided, the conditional probability of dyslipidemia was as depicted in Figs 6C and 6D. For the TG/HDL ratio in Q4, the probability of dyslipidemia was 0.92 when there was no hypertension (Hypertension = "no"), and 0.91 when hypertension was present (Hypertension = "yes"), with only a minimal difference (Fig 6C). For the TyG index in Q4, the conditional probability of dyslipidemia was 0.85 without hypertension (Hypertension = "no") and slightly increased to 0.86 with hypertension (Hypertension = "yes") (Fig 6D).

## Discussion

This study investigated the relationships between IR indicators (HOMA-IR, TyG index, TG/HDL ratio) and their associations with diabetes and dyslipidemia using the Bayesian network analysis. Our findings reveal that the TyG index serves as a comprehensive biomarker capable of predicting both diabetes and dyslipidemia simultaneously, while the HOMA-IR and the TG/HDL ratio are more effective at predicting individual conditions.

The analysis revealed a sequential connection among the three indices structured as TG/HDL ratio→TyG index→HOMA-IR, with the TyG index functioning as a connecting variable (d-separator) between the other two indices. This pattern is particularly informative because a Markov blanket identifies the complete set of variables that directly influence or are influenced by a target variable, effectively filtering out indirect relationships. The presence of TyG index in both HOMA-IR and TG/HDL ratio's Markov blankets indicates that it provides essential information for understanding both measures.

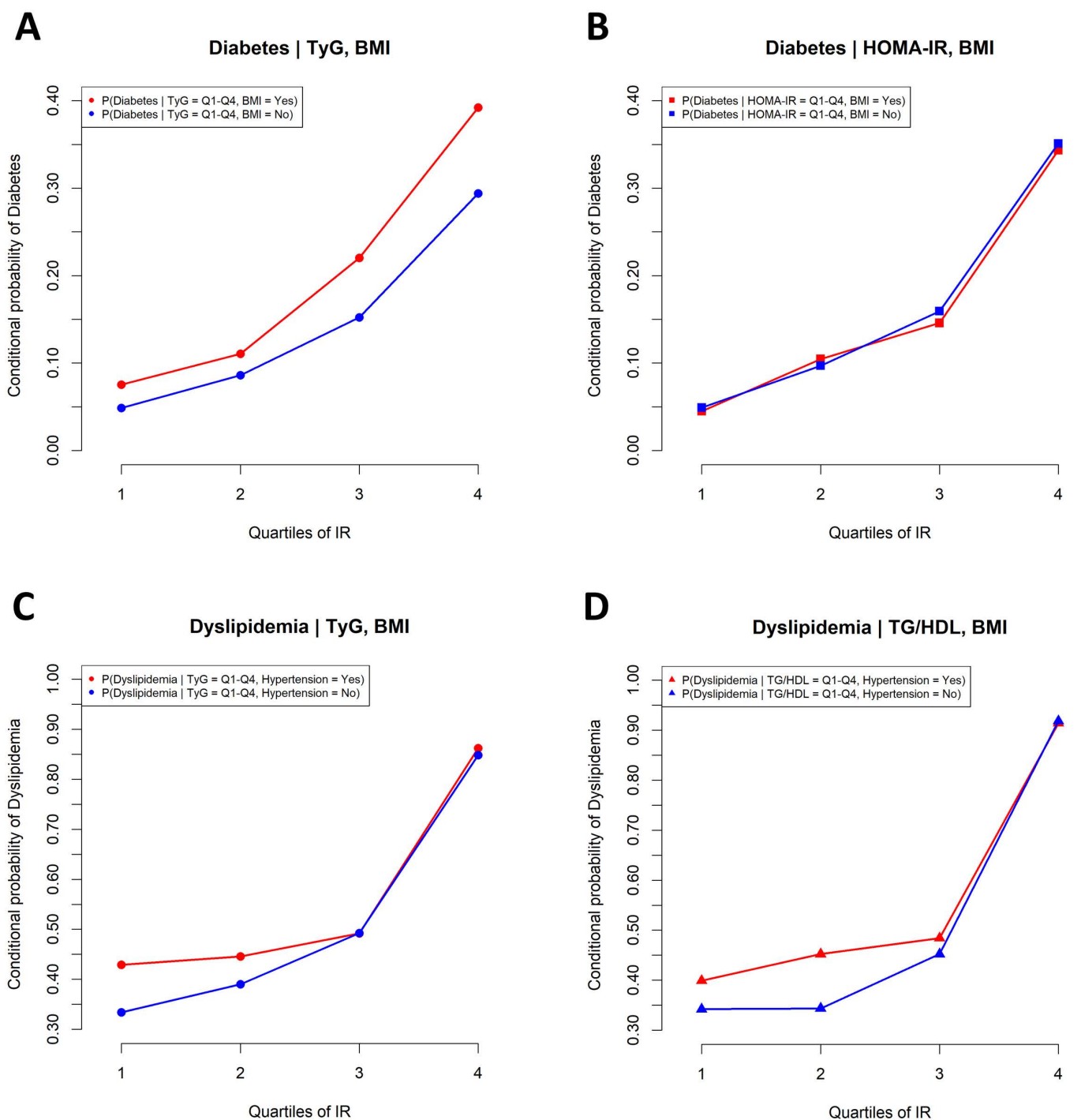

**Fig 6. Conditional probabilities of diabetes with obesity (based on BMI) given the TyG index (A) and HOMA-IR (B), and conditional probabilities of dyslipidemia with hypertension given the TG/HDL ratio (C) and TyG index (D).** Abbreviation: HOMA-IR, homeostasis model assessment of insulin resistance; TyG index, triglyceride-glucose index; TG/HDL, triglyceride-to-high-density lipoprotein cholesterol ratio; BMI, body mass index.

Conversely, the absence of HOMA-IR from TG/HDL ratio's Markov blanket (and vice versa) suggests these two indices represent distinct aspects of IR that become conditionally independent when TyG index is known. This implies that while all three indices measure IR, they capture different physiological dimensions, with the TyG index serving as a central link.

For diabetes prediction, the HOMA-IR and TyG index demonstrated comparable excellence through multiple analytical approaches: both showed high mutual information and AUC values in the ROC curve (Figs 2A and 2B) and exhibited strong predictive power in the Bayesian network conditional probability analysis, consistently outperforming the TG/HDL ratio's performance (Fig 5A). Similarly, in predicting dyslipidemia, the TG/HDL ratio and TyG index showed superior performance across all measures—mutual information, the AUC values, and the Bayesian network conditional probabilities—consistently outperforming the HOMA-IR (Figs 2C, 2D, and 5B).

The strength of connections with FBG, the central variable of diabetes, decreased from the TyG index to the HOMA-IR to the TG/HDL ratio (Fig 3). Similarly, TG, the central variable of dyslipidemia, showed the strongest connection with the TyG index, indicating its robust associations with both glucose and lipid metabolism markers.

The Markov blanket analysis revealed distinct characteristics for each index. The TyG index's Markov blanket included FBG, TG, and HDL, reflecting balanced contributions from glucose and lipid metabolism, along with health behavior variables such as smoking history and drinking frequency (Fig 4C). The HOMA-IR's Markov blanket primarily captured glucose metabolism and obesity characteristics, including FBG and obesity-related variables such as BMI, WC, and WHtR (Fig 4A). The TG/HDL ratio's Markov blanket comprised muscle activity and lipid variables, emphasizing its lipid-centered nature (Fig 4B).

These patterns align with the biological basis of IR: insulin and glucose reflect IR in hepatic cells, while TG and HDL reflect IR in adipose cells [49,50]. The HOMA-IR specifically reflects hepatic IR through insulin and glucose measurements, while the TG/HDL ratio specializes in adipose cell IR through TG and HDL measurements. The TyG index's incorporation of both FBG and TG enables comprehensive measurement of IR in both tissue types, explaining its central position in the DAG structure linking the TG/HDL ratio and HOMA-IR (Fig 3).

Previous research supports our findings. Studies have shown the TyG index's superior performance over that of the HOMA-IR in predicting diabetes and atherosclerosis in type 2 diabetes patients, and better predictive capability than the TG/HDL ratio for diabetes risk in prediabetic individuals [15,17,51]. Che et al. [27] demonstrated the TyG index's greater explanatory power, accounting for 45.8% of dyslipidemia variance and 27.0% of diabetes variance, compared to 40.0% and 11.8%, respectively for the TG/HDL ratio.

When incorporating obesity information, the TyG index showed significant probability value changes based on obesity presence (Fig 6A), while the HOMA-IR showed minimal changes (Fig 6B). This difference arises from the parent–child node relationship between the HOMA-IR and obesity variables, where obesity information is inherently incorporated into the HOMA-IR (Figs 3 and 4A). The TyG index's Markov blanket excluded obesity variables, allowing direct influence of obesity information on diabetes prediction. This finding aligns with studies showing improved predictive effects of combined TyG-obesity indices (TyG-BMI, TyG-WC, TyG-WHtR) compared to the TyG index alone [52–54]. For dyslipidemia, including hypertension information induced minimal probability value changes for both the TyG index and TG/HDL ratio (Figs 6C and 6D), as these indices were parent nodes in dyslipidemia-related clinical variables (Fig 3), already embedding relevant information.

Our findings have significant clinical implications. The TyG index, centrally positioned in our network, offers a cost-effective screening tool using only routine measurements. Its robust associations with both diabetes and dyslipidemia support its use for early risk assessment, potentially improving screening efficiency and resource allocation, especially in resource-limited settings. These insights could enhance early detection and targeted interventions for cardiometabolic diseases.

## Limitations

Our research lacked direct IR validation against actual measurements and excluded sleep duration from the DAG structure despite its presence at lower thresholds (0.55–0.70), as it disappeared at stronger thresholds (0.75+) and showed no

probabilistic influence on key variables. This exclusion likely had minimal impact, though a 2021 format change and our age-diverse sample may have contributed to this pattern. Additional limitations include potential unmeasured confounders such as dietary patterns, occupational factors, and psychiatric conditions that could influence IR pathways—a constraint inherent to observational studies, limited sample size due to recent insulin measurement data unavailability, possible result variation based on hyperparameter settings, and limited generalizability beyond the Korean population.

Finally, it is important to note that our Bayesian network analysis and the resulting DAG do not establish causal relationships between variables. The arrows in our DAG represent probabilistic associations and conditional dependencies learned from the data, rather than proven causal pathways [19]. While these associations provide valuable insights into the relationships between IR markers and metabolic disorders, causal inference would require different methodological approaches, such as interventional studies or causal inference frameworks with additional assumptions [55]. Therefore, our findings should be interpreted as revealing strong statistical associations rather than definitive causal mechanisms. Given that a network structure model was derived in this study, we propose exploring causal relationship inference based on this model as a potential direction for future research.

However, our use of the Bayesian networks offers advantages over traditional regression models by enabling unsupervised learning of probabilistic relationships and providing a visual representation through DAGs. This approach captures complex interrelationships that regression models may overlook.

Furthermore, existing methods for assessing IR and metabolic disorders are often invasive, costly, or complex, limiting their clinical utility. Our study addresses this gap by employing Bayesian networks to identify probabilistic relationships among IR markers and metabolic disorders, offering a novel and more accessible tool for risk assessment in routine clinical practice.

## Conclusions

Diabetes and dyslipidemia are chronic diseases with high comorbidity, significantly increasing cardiovascular risks. While HOMA-IR effectively predicts diabetes and TG/HDL ratio predicts dyslipidemia, both show limited crossover predictive ability. The TyG index bridges this gap by demonstrating excellent predictive performance for both conditions. When combined with obesity information, the TyG index shows enhanced predictive capability compared to HOMA-IR, highlighting its scalability. Additionally, the TyG index's calculation from routine laboratory tests without insulin measurements makes it more cost-effective and practical for clinical use.

## Supporting information

**S1 Fig. A directed acyclic graph of Bayesian network with marginal probabilities.** Abbreviation: BMI, body mass index; WC, waist circumference; WHtR, waist-to-height ratio; SBP, systolic blood pressure; FBG, fasting blood glucose; HbA1c, hemoglobin A1c; TG, triglycerides; HDL, high-density lipoprotein; LDL, low-density lipoprotein; TCHOL, total cholesterol; HOMA-IR, homeostasis model assessment of insulin resistance; TyG index, triglyceride-glucose index; TG/HDL ratio, triglyceride-to-high-density lipoprotein cholesterol ratio.
(TIF)

## Author contributions

**Conceptualization:** Hyun Sook Oh, Jaeyeop Choi.

**Data curation:** Jaeyeop Choi, Jonghyun Kim.

**Formal analysis:** Jaeyeop Choi.

**Investigation:** Hyun Sook Oh, Jaeyeop Choi.

**Methodology:** Hyun Sook Oh, Jaeyeop Choi, Jonghyun Kim.

**Project administration:** Hyun Sook Oh.

**Resources:** Hyun Sook Oh, Jaeyeop Choi.

**Software:** Jaeyeop Choi, Jonghyun Kim.

**Supervision:** Hyun Sook Oh.

**Validation:** Hyun Sook Oh, Jaeyeop Choi, Jonghyun Kim.

**Visualization:** Jaeyeop Choi.

**Writing – original draft:** Hyun Sook Oh, Jaeyeop Choi.

**Writing – review & editing:** Hyun Sook Oh.

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
