## [Decision Letter · Decision Letter 0]

28 Feb 2025

PONE-D-25-06103Relationship between insulin resistance surrogate markers with diabetes and dyslipidemia: A Bayesian network analysis of Korean adultsPLOS ONE

Dear Dr. Oh,

Thank you for submitting your manuscript to PLOS ONE. After careful consideration, we feel that it has merit but does not fully meet PLOS ONE’s publication criteria as it currently stands. Therefore, we invite you to submit a revised version of the manuscript that addresses the points raised during the review process. **Decision: Major revision**

We look forward to receiving your revised manuscript.

Kind regards,

Marwan Al-Nimer

Academic Editor

PLOS ONE

**Journal Requirements:**

Please ensure that your manuscript meets PLOS ONE's style requirements, including those for file naming. The PLOS ONE style templates can be found at https://journals.plos.org/plosone/s/file?id=wjVg/PLOSOne_formatting_sample_main_body.pdf and https://journals.plos.org/plosone/s/file?id=ba62/PLOSOne_formatting_sample_title_authors_affiliations.pdf

**Additional Editor Comments:**

The TG/HDL ratio is an optimal indicator of dyslipidemia. I suggest to add to this ratio the term:"plasma atherogenic index". Add a p-value of <0.05 is a significant level. in the statistical analysis section

Reviewers' comments:

Reviewer's Responses to Questions

**Comments to the Author**

1. Is the manuscript technically sound, and do the data support the conclusions?

Reviewer #1: Partly

Reviewer #2: Yes

2. Has the statistical analysis been performed appropriately and rigorously? 

Reviewer #1: No

Reviewer #2: Yes

3. Have the authors made all data underlying the findings in their manuscript fully available?

Reviewer #1: Yes

Reviewer #2: Yes

4. Is the manuscript presented in an intelligible fashion and written in standard English?

Reviewer #1: Yes

Reviewer #2: Yes

5. Review Comments to the Author

**Reviewer #1: ** Overall

I read with great interest the article entitled “Relationship between insulin resistance surrogate markers with diabetes and dyslipidemia: A Bayesian network analysis of Korean adults”, which falls within the aim of this journal. This study aimed to evaluate the interrelationships among insulin resistance markers (HOMA-IR, TyG index, and TG/HDL ratio) and their associations with diabetes and dyslipidemia using Bayesian network analysis.

In my honest opinion, the topic and results are interesting for audiences and the paper is almost well structured enough to attract the readers’ attention. However, the authors should consider and clarify some points and improve the paper, as suggested below [*: major points, #: minor points].

Abstract section

# Please note that you have considered fasting blood glucose; therefore, it is preferable to use FBG instead of GLU, which is merely an abbreviation for glucose, throughout the text.

# Preferably, replace ‘a distinct serial connection pattern’ with “a sequential association pattern”.

Additionally, use ‘established association with’, instead of “inherent incorporation of”.

Moreover, instead of “exceptional” in “exceptional tool”, which is not a commonly used term in articles, use like ‘effective’.

Material and methods section

# Preferably, provide a brief explanation alongside the phrase “participants outside the age range of 40–70 years” to clarify the reason for this selection. For example, you did not consider individuals outside this age range because the risk of diabetes and dyslipidemia is not high in those groups.

* In the Study Design and Data Extraction part, you have referenced [27,28] as well as [28] separately. Could you please clarify exactly what these references pertain to and the purpose of citing them?

# Preferably, include a mention of the participants’ fasting status wherever you discuss the measurement of indices.

# Preferably, cite the relevant references after the formulas for the TyG index and TG/HDL ratio.

* Please provide some details on how household income was categorized and cite an appropriate reference(s) for it. Additionally, include suitable references for the categorization of sleep and smoking status.

* In the Statistical Analysis part, please specify the comparisons made between the two groups. Additionally, include an explanation of the significance threshold, clarifying the interpretation of P-value < 0.05.

# Preferably, provide a brief explanation in the text regarding why Rao-Scott chi-square tests were used, instead of Pearson’s Chi-Square Test.

* Since you mentioned that you compared the accuracy of the three markers (HOMA-IR, TyG index, and TG/HDL ratio), please ensure that the interpretations in the following section explicitly incorporate the term accuracy.

Results section

* Please ensure that throughout the text and in all tables, the number is presented alongside the percentage, following the format n (%).

* Instead of directly referring to Q1 to Q4, please provide an interpretation of their meaning.

Additionally, preferably include an interpretation of their statistical significance. I think that it should be sufficient to mention which group significantly had the highest quartile (Q).

* Before interpreting the structure of Directed Acyclic Graph (DAG) in this section, please first clarify the purpose of using that in the previous section.

* Additionally, it seems likely that you drew the DAG manually rather than using the online DAGitty version 3.1 software. Preferably, recreate the relationships using the software and upload the updated image. This will allow you to identify and correct any cyclic models by adding or removing specific relationships. Moreover, some relationships currently appear incorrectly drawn. For example, education does not influence gender; rather, the reverse is true. Similarly, the relationships between muscle activity and TG/HDL with gender seem misrepresented. Please review and adjust these accordingly.

* Preferably, provide an interpretation of the Markov blanket in Figure 4, making it as clear and intuitive as possible to ensure easier understanding for readers with limited familiarity with the concept. Additionally, briefly mention the purpose of using the Markov blanket in the previous section to provide better context.

* Regarding the sentence “Hypertension was not part of the Markov blanket of any of the three indices; however, it showed the strongest association with the TG/HDL ratio”, please clarify the second part of the sentence. Currently, in Figure 3, there is no line indicating a relationship between them, so I personally do not understand the association between hypertension and the TG/HDL ratio.

Good Luck,

**Reviewer #2: ** This manuscript presents a comprehensive analysis of insulin resistance (IR) surrogate markers—specifically the Homeostatic Model Assessment of IR (HOMA-IR), the TG-Glucose (TyG) index, and the TG-to-HDL ratio (TG/HDL ratio)—using data from the Korean National Health and Nutrition Examination Survey (2019–2021). The authors employ Bayesian network analysis to explore the relationships between these markers and their associations with diabetes and dyslipidemia. The study identifies a distinct serial connection pattern among the markers, positioning the TyG index as a central connecting variable. The findings suggest that while HOMA-IR is effective for predicting diabetes and the TG/HDL ratio is more suited for assessing dyslipidemia, the TyG index demonstrates robust predictive capabilities for both conditions, particularly when integrated with obesity information. The analysis also highlights the unique metabolic signatures of each marker, contributing valuable insights into their clinical utility.

Comments and Suggestions for Improvement:

1. Uniformity in Decimal Usage: The manuscript should maintain a consistent format for decimal representation throughout. For example, please standardize the use of either one decimal place (e.g., .0) or two decimal places (e.g., .00) across the entire document.

2. Abbreviation Table: It is recommended to include a table of abbreviations. While abbreviations should be defined upon first use in the text, a dedicated table will provide readers with a quick reference, enhancing clarity and understanding of the terms used throughout the study.

3. Citing Figures and Tables: Ensure that figures and tables are appropriately cited within the results section. Additionally, the discussion section should include interpretations of these figures and tables to provide context and enhance the reader's understanding of the findings.

4. Future Research Directions: In the limitations section, please provide a brief discussion on potential future research directions that could address the identified limitations. This would offer valuable insights into how subsequent studies might build upon your findings.

5. Public Health Implications: The discussion section currently lacks a thorough exploration of the public health implications of your findings. It would be beneficial to elaborate on how these results could influence public health strategies and clinical practices related to diabetes and dyslipidemia management.

6. Connection Between Limitations and Objectives: A more explicit connection between the limitations of current methods and your study's objectives would strengthen the rationale for your research. This would help clarify why the study is necessary and how it contributes to the existing body of knowledge.

7. Streamlining Redundant Information: Some sections of the manuscript contain redundant information that could be streamlined to maintain reader interest and focus. For instance, the detailed explanation of hyperglycemia mechanisms could be condensed to allow for a more direct presentation of the study's aims.

8. Presentation of Quartile Data: The presentation of quartile data for the HOMA-IR, TyG index, and TG/HDL ratio is overly detailed. The extensive listing of percentages across quartiles may overwhelm readers and obscure the main findings. Summarizing these results in a more concise manner, potentially through visual aids or summary statistics, would improve clarity and focus.

9. Exclusion of Sleep Duration and Confounders: You mention the exclusion of sleep duration from the Directed Acyclic Graph (DAG) structure but do not explain how this exclusion may have impacted the results. Additionally, the mention of "potential unmeasured confounders" is too general; please specify what these confounders might be and how they could influence the study's findings.

6. PLOS authors have the option to publish the peer review history of their article (what does this mean? ). If published, this will include your full peer review and any attached files.

**Do you want your identity to be public for this peer review?** For information about this choice, including consent withdrawal, please see our Privacy Policy .

Reviewer #1: No

Reviewer #2: No

---

## [Author Response · Author response to Decision Letter 1]

2 Apr 2025

We sincerely appreciate the opportunity to revise and resubmit our manuscript, titled “Relationship between insulin resistance surrogate markers with diabetes and dyslipidemia: A Bayesian network analysis of Korean adults”, for consideration in PLOS ONE. We are grateful for the constructive feedback provided by the reviewers and the editor, which has greatly helped us refine our work.

In response to the reviewers' comments, we have carefully revised the manuscript and provided a detailed, point-by-point response in a separate document. We have carefully addressed all concerns and believe that these revisions have strengthened our manuscript. We have also uploaded the revised manuscript with tracked changes and a clean version reflecting all modifications.

We sincerely appreciate your time and consideration of our revised submission.

---

## [Editor Report · Decision Letter 1]

6 Apr 2025

Relationship between insulin resistance surrogate markers with diabetes and dyslipidemia: A Bayesian network analysis of Korean adults

PONE-D-25-06103R1

Dear Dr. Hyun Sook Oh,

We’re pleased to inform you that your manuscript has been judged scientifically suitable for publication and will be formally accepted for publication once it meets all outstanding technical requirements.

Kind regards,

Marwan Al-Nimer

Academic Editor

PLOS ONE

Additional Editor Comments (optional):

None
---

## [Editor Report · Acceptance letter]

PONE-D-25-06103R1

PLOS ONE

Dear Dr. Oh,

I'm pleased to inform you that your manuscript has been deemed suitable for publication in PLOS ONE. Congratulations! Your manuscript is now being handed over to our production team.

Kind regards,

on behalf of

Dr. Marwan Al-Nimer

Academic Editor

PLOS ONE